# The Development and Pilot Evaluation of a Family-Based Education to Strengthen Latinx Adolescent Mental Health in the United States: The *Familias Activas* Experience

**DOI:** 10.3390/ijerph20010207

**Published:** 2022-12-23

**Authors:** Ghaffar Ali Hurtado Choque, Hilda Patricia García Cosavalente, Alexander E. Chan, Matthew R. Rodriguez, Eva Sumano

**Affiliations:** 1Department of Family Science, School of Public Health, University of Maryland, College Park, MD 20742, USA; 2College of Agriculture and Natural Resources, Extension, University of Maryland, College Park, MD 20742, USA; 3University of California Cooperative Extension, Auburn, CA 95603, USA; 4CASA de Maryland, Adelphi, MD 20783, USA

**Keywords:** Latinx youth, mental health, positive youth development, sports

## Abstract

Adolescent years are a time of joy and can represent a challenge for parents and youth, especially for immigrants to the US who are adjusting to their host country. Programs focusing on family skills and positive youth development (PYD) can contribute to youth wellbeing especially, however, few exist for low-income immigrant families. (1) Background: The major goals of this project are to strengthen both PYD and healthy parenting practices by implementing an evidence-informed program, *Familias Activas*. A theory of change guided the development of *Familias Activas* in which three factors: parent training, positive youth development, and youth physical activity sessions (soccer) aimed to improve Latinx youth mental health. Youth participated in weekly soccer practices led by trained soccer coaches while parents/caregivers attended parent education. Both sessions lasted eight weeks. (2) Method: We describe the formative stage of the research project as well as the pilot implementation of the *Familias Activas* program, which provides critical insights for the development of a PYD program. (3) Results: Evaluation surveys were administered to youth and their parents. Thirty youths and sixteen parents completed the survey. The Kidscreen scale had a mean for most items ranging from 3.6 to 4.2. Participating youth were 11 years old and most affirmed they were Latinx. The feasibility program quality mean was 4.2 indicating an overall positive result for the pilot program.. Implications of PYD programs for Latinx youth are discussed. (4) Conclusions: The current paper presents a model for positively influencing the physical and mental wellbeing of Latinx youth and their parents. The model is culturally responsive in its involvement of both parents and youth in programming.

## 1. Introduction

Adolescent years can represent a challenge for both parents and youth, and especially for immigrants or the so-called “Generation 1.5,” who are adjusting to their host country or learning new cultural values. Identity processes affect youth and parents as they embark on their immigration journeys. Adolescents face many challenges while they aim to explore their identities and remain close to their original and host countries [1]. Research shows that both racial and ethnic disparities regarding health still prevail among youth. Moreover, it is not uncommon for Latinx youth to face poorer mental health outcomes [2] as shown in the systematic review of socioecological factors, community violence exposure, and disparities for Latinx Youth; even before the pandemic, studies of Latinx youth showed high rates of mental health concerns [3,4]. Some of these negative outcomes may be attributable to unique stressors among this population, such as immigration political rhetoric fueling deportation fears. Research has shown that deportation fears increase the risk for depression in both youth and parents of Latinx families [5]. In addition to unique minority stress influences, prior research suggests disparities in service utilization as an explanation for poorer mental health outcomes among Latinx youth. Latinx youth are less likely to receive mental health care compared with non-Hispanic White youth [6]. Parents’ level of acculturation may contribute to differences in receipt of care [7]. Furthermore, parents of youth with mental health needs often report facing stigma and a lack of information about recognizing mental health concerns as a barrier to achieving optimal mental wellness for their children [8].

Moreover, nearly half of America’s youth population is becoming more racially diverse. Almost half of the youth population (10–19 years old) identified as coming from an ethnically diverse group, such as 24.8% of the American youth identifying as Hispanic [9]. To remain relevant, public health programs need to adapt to this demographic shift among youth by offering culturally sensitive and convenient positive youth development programs. As public health specialists, it is critical to provide resources and tools to immigrant youth and their families on how to successfully navigate the adolescent years.

The following sections of this paper will detail positive youth development (PYD) principles, how a PYD project to address the mental and physical wellbeing of urban Latinx youth was developed, and some initial results of the program. Positive youth development programs can contribute to improve the development of youth, especially those at risk, such as immigrants, refugees, and youth from low-income families. The major goals of the project were to strengthen both the PYD of Latinx youth and healthy parenting practices by implementing a new program called *Familias Activas*. There is still a need to further knowledge regarding the research and evaluation of PYD programs that aim to improve Latinx youth mental health. This unique program aims to improve youth mental health using soccer sessions and the simultaneous improvement of parenting skills and attitudes. Thus, the program theory that guided the development of *Familias Activas* included three interrelated components: parent training, positive youth development, and youth physical activity aiming to improve Latinx youth mental health.

### 1.1. Positive Youth Development

In the 1990s, a youth development framework emerged in the US as a set of principles and philosophy that stresses a support for youth development. This approach paved the way for the design of programs, projects, and activities that aimed to foster youth development [10]. Youth development programs emerged in different realms, such as education and public health. Public health prevention programs focusing on adolescent health tend to have a deficit approach by focusing on risk factors and youth health problems; however, the positive youth development approach challenges this deficit approach by focusing on the positive characteristics of youth [11]. Damon [12] defined positive youth development as the focus on the potential of youth rather than on their supposed incapacities. This approach aligns with new projects and programs sponsored within societies and communities of color.

The evaluation of PYD programs suggests that this type of approach provides multiple benefits to youth participants. One of the benefits of a PYD approach is that it can improve youth’s mental health and well-being and improve critical skills such as resilience and self-efficacy [13]. Furthermore, previous research has found that youth who participate in sports-based interventions gain social skills such as leadership, goal setting, problem solving, and time management [14] and can decrease negative outcomes such as illegal behavior among high-risk Latinx and African American youth in the US [15].

In the US, PYD approaches are often incorporated into sports programs. For example, a PYD program through basketball in the western region of the US showed that structured sports programs can promote positive character development, such as autonomy [16]. Bates, Greene, and O’Quinn [17] found that even virtual sports-based PYD programs in Los Angeles, CA during the COVID-19 pandemic were able to engage Latinx youth and their families, improving youths’ life skills. Other research showed positive results of a sports youth development program for Latinx youth regarding significant improvement in social skills, school connectedness, and peer support [18]. PYD programs in New York City [19] showed the barriers low-income urban youth face in their neighborhoods that prevented them from engaging in physical programs, such as violence exposure and lack of resources, or to engage in structured-sports programs, shedding light on how to engage vulnerable youth in PYD programs.

Participation of Latinx families in PYD can be beneficial for many reasons. It is documented that Latinx youth face health challenges, as detailed earlier, and with effective sports-based programs targeted to Latinx youth, they can develop life skills that will give them support throughout their adult lives. For example, sports programs help youth develop the “5 C’s” of competence, confidence, character, connection, and caring [20]. Youth develop skills (competence), which are encouraged by caring adult coaches (confidence, connection), are taught sportsmanship (character), and are encouraged to support one another during practice (caring). Taken together, extant research supports the use of PYD approaches in community-based youth programming. Although typical sports programs may not incorporate PYD principles, emerging research suggests that sports can be an effective vehicle for applying PYD approaches. Sports-based programs provide youth with the attention of caring adults, increase youths’ self-efficacy, provide exercise, and can also contribute to character development. These individual and community characteristics are linked to resilience in the face of adversity. When developing and implanting PYD youth programs, it is important to consider the specific cultural needs and strengths of Latinx families. Cultural needs are related to offering public health services in their preferred languages, considering the involvement of multiple family members in promoting health, and adapting youth programs to their specific health needs. Latinx family strengths are related to the protective factors of biculturalism and familism. Biculturalism is defined as a notion that an individual has internalized two different cultures and switches between them in their daily interactions [21]. There is research that suggests that bicultural youth have positive assets that help them navigate and adapt to different environments [22]. Additionally, familism has been shown to offer protective factors to Latinx youth [23] such as family cohesion and less parent–child conflict [24]. Familism has also been shown to buffer young males against risky situations such as violence or drug use [22].

As shown in this section, it is critical to address the mental health concerns of Latinx youth in the US. Tailoring an engaging sports-based project for Latinx youth and their families is critical to advancing health equity. The *Familias Activas* project aimed to improve the mental health of youth as well as to provide family education to parents’ youth. The next sections provide details on the development and implementation of the *Familias Activas* pilot project.

### 1.2. Study Objective

*Familias Activas* is a federally funded (Children Youth and Families at Risk, CYFAR-USDA) community-based participatory research project that aims to strengthen youth mental health and lower youth anxiety among Latino youth (ages 10–14) by engaging families (especially fathers) in a multi-component program including: (1) culturally and linguistically appropriate family life education, (2) positive youth development programming (PYD) focusing on mental health, and (3) youth physical activity sessions (soccer practices). The program is being developed guided by the ecodevelopment and social cognitive theories [25,26], and following principles of community-based participatory research (CBPR) in partnership with Latino families and trusted community-based organizations serving Latinx families in Maryland, USA, and aims to build on the strengths of immigrant Latino families.

This program is led by the School of Public Health and the University of Maryland Extension. The first component, (1) the culturally and linguistically appropriate family life education, is part of the *Padres Preparados*, *Jóvenes Saludables* (Prepared Parents, Healthy Youth) [27] curriculum that consists of eight educational sessions for parents, which includes topics such as healthy parenting practices, communication, conflict management, developing parent–youth connection, parenting across cultures, and nutrition and physical activity. “*Padres Preparados*” engages Latinx youth and their families, given the need for more culturally informed evidence-based programs that includes parents and their children. In 2021, the current project ‘*Familias Activas*’ added the soccer component to the project to engage youth. Both projects are related in the sense that they both engage Latinx families and at-risk youth. The second component, (2) positive youth development programming (PYD) that focuses on mental health was implemented during the soccer sessions. Soccer coaches use positive affirmation strategies that aim to enhance youth mental health. Having these trusted adults as coaches also aims to give youth a chance to express their emotions and feelings in a safe environment. The third component is (3) youth physical activity sessions (soccer practices). Soccer is a popular sport among Latinx youth in the US. During soccer sessions, youth practice skills such as teamwork, communication, and conflict management.

Considering that the main long-term goal is to improve youth mental health, we hypothesized pathways of change regarding youth’s mental health and physical activity. As shown in Figure 1, through the three components (parent training, positive youth development, and youth physical activity) youth mental health could be improved.

In addition, short-term desired results were developed, as follows: (1) youth will increase physical activity because of participating in the program, (2) youth will increase their mental wellbeing because of participating in the program, and (3) parents will be engaged, providing family life education.

## 2. Materials and Methods

### 2.1. Study Design and Contextualization

A collaboration among researchers at the University of Maryland College Park, community leaders at Casa de Maryland, and Kick-Like-A-Girl (soccer coaches concerned with healthy youth development) was established. These organizations identified core cultural values, community assets, and health priorities. Internally, our core research team, which included researchers and mental health specialists from the University of Maryland, engaged several students (graduate and undergraduate) to assist with the program.

We have also engaged: (1) a social media coordinator, (2) a program coordinator, and (3) a technology specialist; and hired two cooperative extension staff: (1) a mental health specialist and evaluator and (2) a 4-H positive youth development educator. 4-H refers to extension programs offered by 100 public universities that aim to engage youth across the US.

The collaborative team (including university and community partners) discussed how to integrate youth, parent, and community concerns including mental health topics, physical activity, and family education strategies with theoretical behavior change models. The pilot program took place in a diverse community in Maryland in the spring of 2022.

The parent training component comes from a previous successful experience called *Padres Preparados*, *Jovenes Informados* which has shown to be feasible in preventing negative health outcomes of Latinx youth in the US [28]. These positive results have guided our pilot project. This family skills training intervention, based on a social ecology model, has shown positive results including improved parent involvement, child social competence, and child self-regulation. Each educational session is divided between self-reflection, didactics, and skill-building exercises, all aimed at developing strong parenting practices and facilitating relationship-building between parents and youth. These educational sessions were given to parents while their kids participated in the soccer sessions.

The mental health and sports (soccer) portions of the program aim to engage youth in physical activity. This physical activity portion was led by adults trained in mental health. This model is built on a plan that is based on the importance of connectedness, good communication, and guidance. Moreover, these sports programs are intentionally designed to be accessible to all youth. The mental health portion was given during the soccer sessions. Coaches worked on motivating youth to do their best, improve their self-perception during the games, and improve their social skills with their peers.

### 2.2. Key Personnel, Participants, and Recruitment

We connected with three experienced soccer coaches, who are part of the Kick-Like-A-Girl organization and worked directly with the youth. These soccer coaches have extensive experience with the game of soccer. The program delivery team met on a weekly basis to review our program goals and plan for how they would engage with youth regarding both the soccer and mental health components. All the soccer coaches participated in a training program entitled: “Youth Mental Health First Aid.” Our community partner, Casa de Maryland, also hired a coordinator who became part of the core research team. This staff person had a background in mental health and soccer. In addition, Casa provided three additional staff to assist with our recruitment efforts through their community *promotores* (community health workers). Through their efforts, we recruited thirty families. These families were immigrant Latinx families whose primary language was Spanish. Most of the parents had low levels of education and income. The average age of parents was 40 years old, and the average age of the youth was 12 years old.

### 2.3. Community Engagement

To ensure the success of the project, it was critical to engage families of Latinx youth as well as the community; for this reason, we hosted a community event at the program site three months in advance of the program start date. Approximately 15 people attended. This community event was an important part of the program’s attempt to establish trust and rapport with the community so that the community would be familiar with the program and its staff, who were not all closely integrated in the community at the start of the program.

### 2.4. Website and App Development

At the end of 2021 and before the pilot was launched, we successfully designed, developed, and published a custom website at https://familiasactivas.org (accessed on 21 November 2022) along with two mobile applications offered for free download in the iTunes and the Google Play stores (See Figure 2). The website and apps were developed specifically for the “*Familias Activas*” project. The goal of creating a website was to increase our communications with the Latinx community and it was therefore translated to Spanish to reduce language literacy barriers. To increase sustainability, we trained several staff on how to manage the content on the website. The mobile app provided a mechanism to deliver the program curriculum to our participants, such as mental health and soccer training videos. The app’s text messaging functionality allowed the program staff to text youth and parents with encouraging messages and reminders about upcoming program sessions. As part of the pilot program, we also provided custom-designed soccer jerseys for soccer coaches and staff (See Figure 3).

### 2.5. Structure of Youth and Parent Sessions

Youth sessions: the program met weekly on Saturdays for one hour (each session) during eight weeks for youth to play soccer while learning social skills and healthy mental health habits. For example, soccer coaches began each practice with a mindfulness-based intention-setting activity [29], whereby youth thought of a positive word or phrase that would guide them through practice. The soccer program included structured activities such as scrimmage and games. Soccer was selected as the main sport because it is not only a popular sport among Latinx youth but also soccer activity interventions showed the strongest evidence for improvements in physical self-perceptions among youth [30].

Parents’ sessions: the program met weekly on Saturdays and these sessions were run parallel to the youth sessions. This timing was convenient because parents dropped off their youth at the soccer sessions and then headed to the parents’ session. The approach to parenting practices was strengths-based and was built on culturally grounded parenting assets and not from a deficit model. Despite multiple stressors in their lives, parents are resilient. We believe that practices that are culturally grounded such as the importance of respect, relationships, and family support [31] are protective and foster healthy development. This also relates to parents/caregivers with high levels of acceptance, attachment, and other protective capacities. Family skills programs can boost parents’ confidence and social support. The topics that were covered during the parents’ sessions are included in Table 1.

While healthy parenting practices and parent–youth relationships are protective, events such as the pandemic can disrupt family functioning and undermine parenting processes [32,33]. These events exacerbated economic stressors, illness, and housing instability, and could generate distress in families. The added stress can affect their family wellbeing and increase parental depression and anxiety [34]. Family-focused programs during COVID-19 need to consider these added stressors within their work to support healthy youth development. Because adolescents are likely to receive mental health services in schools, compared to other community settings that were disrupted due to COVID-19, we are filling this gap by engaging family support.

### 2.6. Measures

#### 2.6.1. Youth

Program quality: the scale proposed by Borden and Perkins [35] had 24 items. There are 5-point Likert scales from 1 = never to 5 = always. Statements were related to how the program may work. Some statements were: “Young people feel safe when they are at the program”. “Young people spread rumors about others”. “Young people keep others from being part of the activities or groups”. “The program has rules about what sorts of behavior are expected”. “Young people contribute to the community by helping others”.

Kidscreen for youth: the scale proposed by Ravens-Sieberer and colleagues [36] included 10 items. There are 5-point Likert scales from 1 = not at all, to 5 = extremely. The statements were questions related to the youth’s wellbeing: “Have you physically felt fit and well?” “Have you felt full of energy?” “Have you felt sad?” “Have you felt lonely?” “Have you had enough time for yourself?”

Resilience: the original scale proposed by Ungar [37] originally had 58 items but we used the modified scale proposed by Jefferies, McGarrigle, and Ungar [38]. This scale had 17 items. There are 5-point Likert scales (1 = not at all, to 5 = a lot). Statements were related to: “I feel supported by my friends” and “I feel that I belong/belonged at my school,” among others.

#### 2.6.2. Parents

Kidscreen for parents: this scale included 10 items [36]. There are 5-point Likert scales from 1 = not at all, to 5 = extremely. The statements were questions related to their youth’s wellbeing: “Have you physically felt fit and well?” “Have you felt full of energy?” “Have you felt sad?” “Have you felt lonely?” “Have you had enough time for yourself?”

Parental acceptance scale: eight items derived from the children’s report of parents’ behavior inventory [39] were included in this scale. There are 5-point Likert scales from 1 = not good at all, to 5 = very good. Some statements were as follows: “I told or showed my child that I liked (him/her) just the way (he/she) was”. “I had a good time with my child.” “I saw my child’s good points more than (his/her) fault”.

Parent personal involvement: five items adapted from Stattin and Kerr [40] were included in this scale. There are 5-point Likert scales from 1 = almost never, to 5 = very good. Statements included: “I spent time with [TC] or did things with him/her alone”. “I started conversations with [TC] about his/her free time activities”. “I started conversations with [TC] about things that happened to him/her during the day”.

After-session evaluation: five items were included in this scale which ranges from 1 = not at all, to 5 = a lot. Statements were related to: “The things I learned today are helpful to me as a parent”. “I felt comfortable sharing my opinions”. “The sessions held my interest”. “The facilitator provides support that helps my learning”. “The facilitator addresses the needs and interests of the group”.

#### 2.6.3. Reliability of Measures

Program quality: the 24 items were scored, and the Cronbach score was 0.820. The items were related to how the program would work and how youth evaluate it.

Kidscreen: the 10 items for this scale obtained a Cronbach score of 0.788.

Kidscreen (parents): the Cronbach score couldn’t be obtained because there was not enough data to measure the reliability of the scale.

Resilience: the 17 items of resilience were scored, however 7 items had variables with zero variance, so SPSS deleted them. The scale included 10 items and the Cronbach score was 0.790.

Parental acceptance scale: 8 items. The Cronbach score couldn’t be obtained because the variance in several items was zero.

Parent personal involvement: 5 items. The Cronbach score couldn’t be obtained because the variance in several items was zero.

## 3. Results

Thirty families participated in the pilot study. While youth participated in the soccer sessions, parents attended the educational program and learned relevant topics related to positive parenting. Surveys were conducted during the pilot and the following section includes the main findings related to youth and parents.

### 3.1. Youth Sessions

Thirty youths completed the survey. Of these, 69% identified as male (N = 20) and 31% identified as female (N = 9). One youth did not provide data for the gender item. Most of the participants were in 5th grade (N = 7), followed by 4th (N = 6), 3rd (N = 5), and 6th grade (N = 5).

The mean age for the youth group was 11 years old; 83% affirmed they were Hispanic/Latino (N = 25). As is typical when studying Latinx groups, most of the participants (N = 20, 66.7%) did not provide a response when asked to indicate their race, but 20% (N = 6) affirmed they were American Indian or Alaska Natives, and 10% (N = 3) affirmed they were white. When referring to their family’s military status, most of them (70%) affirmed their parents were not involved in the military.

Program quality (PQ-Y): a large portion of respondents did not finish this section near the end (question 24). The total mean for this scale was 4.2 which was a positive result for the pilot program overall (1 = never, 5 = always).

Kidscreen scale: only one third of the respondents answered all the questions in this section. The mean for most items ranges from 4.2 to 3.6 (1 = never, 5 = always) which suggests that most youths had positive experiences the last week. The items KS-Y3 [Have you felt sad] and KS-Y4 [Have you felt lonely] differ from the others; these two questions obtained means of 1.5 and 1.6, respectively, which suggest that youth affirmed that they had not experienced sadness and loneliness the past week.

### 3.2. Parents Sessions

The survey for parents was responded to by 16 individuals. The following tables include the demographics items of the survey. Most of the sample was feminine (87.5%), and none was masculine. There was 12.5% missing data regarding this item. Regarding the age variable, the mean age for parents was 38.7 years old. Most of the sample identified as Hispanic/Latino (87.5%). For the race category, 18.8% affirmed they were American Indian or Alaska Native and 12.5% affirmed they were White, however, 68.8% did not complete this question. Concerning the variable regarding level of education, ten individuals answered this question and most of them (56.3%) affirmed they had less than a high school educational attainment. For the question asking if they have ever served in the military including guard or reserves, none of the respondents affirmed they have served in the military.

The average number of sessions participants attended was 6.2 and the average number of hours they participated in each session was 4 h. One question requested respondents to affirm if they had participated in 4-H programs (volunteering youth development programs) and 75% affirmed they had participated in 4-H for less than a year. The responses for the question of whether they participated in other community or volunteer activities had mixed results. Half of the responses (43.8%) indicated they did not participate in these activities and the other half affirmed they did. Missing data was 12.5%.

In the Kidscreen section, most of the data was missing. Only one or two responses were obtained. The mean for the obtained data was mixed because for most items the mean and median were 5 (1 = not at all, 5 = extremely), however, there was one response which skewed the data to the middle (mean = 2.5) for items KS-P3 [Has your child felt sad] and KS-P4 [Has your child felt lonely].

In the Kidscreen, the parental acceptance scale and parent personal involvement scales had to be dropped because internal consistency (Cronbach) could not be obtained.

Program evaluation: process evaluation measures included a parent’s report of session appropriateness (interests, relevance, comfort level, and satisfaction) in the form of an after-session evaluation. The mean for the four items was 3.9 which was considered a high number. Thus, overall, the session was relevant and interesting for parents.

## 4. Discussion

Our pilot study had one main goal, focused on designing and implementing the soccer sessions for youth and educational sessions for parents. The main goal of the project was to strengthen youth mental health and lower youth anxiety amongst Latino youth (ages 10–14). First, we worked on the design of the project. Having partnerships with local organizations was key to assuring the sustainability of the project. Our community partners Casa de Maryland and Kick-Like-A-Girl helped us develop stronger relationships with the Latino community, which allowed us to make the program sustainable.

Physical activities such as soccer are cultural and social activities. In systematic reviews [28], community-based soccer activity interventions were assessed and found to have the strongest evidence for improvements in physical self-perceptions, which accompanied enhanced self-esteem among youth. Sports are linked to psychosocial (social connectedness, mood, and emotions) and behavioral (sleep quality, coping, and self-regulation skills) mechanisms, leading to improved mental health outcomes such as self-esteem, quality of life, and resilience and in the long-term decreased internalizing disorders (including anxiety and depression). The pilot showed that youth were very interested in soccer training and gave overall positive scores in the program quality scale.

Sessions for youth (See Figure 4) and parents (See Figure 5) were well attended, however, some scales of the parent surveys, such as Kidscreen and parent personal involvement, had a large amount of missing data. This could be related to the large number of items in each scale and to the low levels of literacy of our participating parents.

This study has notable strengths and limitations. In terms of strengths, the program model is innovative in its approach to positive youth development. The program simultaneously targets multiple ecological levels of resilience. These include youth factors (e.g., exercise, skill building), familial factors (e.g., parent education), and community factors (e.g., involvement with coaches and non-family adult mentors). Furthermore, this model was applied to a relatively low-resource, ethnic minority population. By including both parents and youth, the program addresses more of the entire family system than programs that focus on only parents or only youth.

The study is limited by the breadth of data collected during initial surveys. Although the investigators initially planned to use more instruments, families and youth were not adequately prepared to spend the time necessary for a more robust measurement of program outcomes. This challenge is being addressed during current and future implementations of this project. Fortunately, the program benefits from a five-year funding cycle, which will allow for the refinement of data collection procedures. The authors also acknowledge personal involvement in the delivery of the program. Therefore, the authors possess an interest in publicizing the program’s success.

The feedback obtained in this pilot program will be used to improve its next phase. For instance, assistance during pre- and post-tests would be provided to youth and parents when filling out the questionnaires. In addition, multiple days for filling out the surveys will be planned due to the several items included in the scales. Assisting parents and youth would improve the rate of response to the surveys.

Findings from the formative pilot study have several implications for public health. Considering that youth mental health was highly impacted by the pandemic, where anxiety and depression rates increased, implementing and developing PYD programs was paramount during this uncertain time. The pilot program provided important lessons that can inform future PYD sports-based programs targeted at Latinx youth. First, the importance of establishing local partnerships with community-based organizations can ease the ownership and sustainability of the program within the community. It is well known that projects that stress the importance of local partnerships will generate effective outcomes. Second, pilot findings provide evidence that a PYD soccer program in a Latinx community is highly beneficial and appropriate for the target population. During the pilot, we ran the sessions at full capacity and had a waiting list of interested participants. Moreover, our results suggested that youth improved their soccer skills during the sessions. Our study is unique in that it encourages parents and youth to participate together with the support of professional soccer coaches as well as experienced public health educators and mental health specialists. A final implication would be related to the unique characteristics of the targeted population: Latinx youth. Most of the participants affirmed they were Hispanic/Latinx, and most parents had lower levels of education attainment and socioeconomic status. This group of families represents a critical group of participants who might be vulnerable to risky behaviors because of the lack of economic resources. PYD programs represent a critical component of development for these youth.

For future directions, and as mentioned above, we currently have 39 youth potentially interested. We are also working with our Casa community organizers “*promotores*” to identify families through their existing WhatsApp communication channels. Our website and free mobile health apps (iTunes and Google Play) in English and Spanish will help the programs establish a necessary system that will increase the long-term sustainability of the *Familias Activas* program.

## 5. Conclusions

The current paper presents a model for positively influencing the physical and mental wellbeing of Latinx youth and their parents. The model is culturally responsive in its involvement of both parents and youth in programming. Both parents and youth reported meaningful gains in the program’s targeted skills and attitudes. Although the evaluation was simpler than originally planned, our pilot implementation has provided valuable formative feedback that will influence the ongoing development of this program. As the community becomes familiar with this *Familias Activas* identity, we seek to build a trustworthy program that is recognizable in the community for providing outstanding opportunities for positive youth development in the Latinx community.

## Figures and Tables

**Figure 1 ijerph-20-00207-f001:**
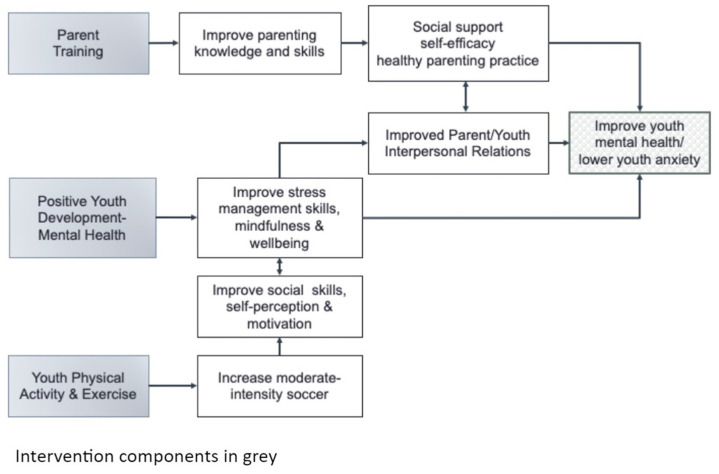
Conceptual model.

**Figure 2 ijerph-20-00207-f002:**
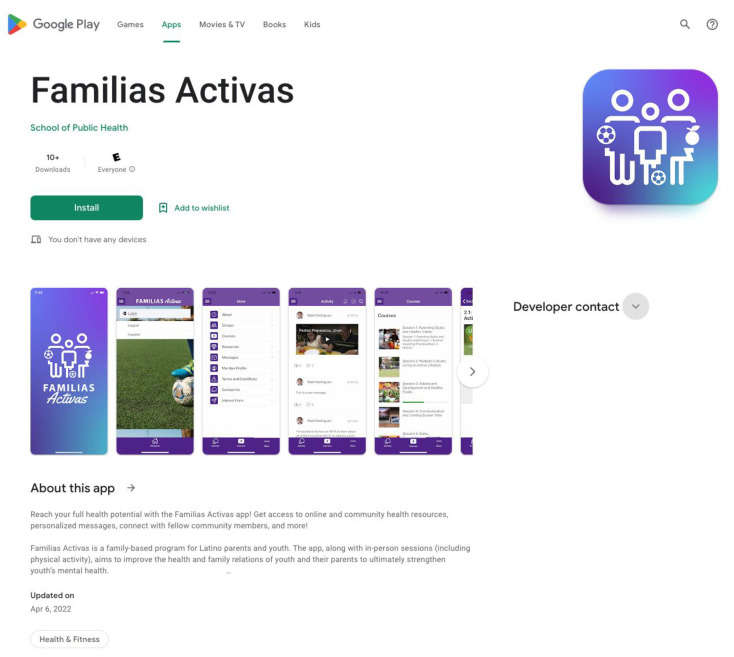
Familias Activas mobile health app (offered in both Google Play and iTunes).

**Figure 3 ijerph-20-00207-f003:**
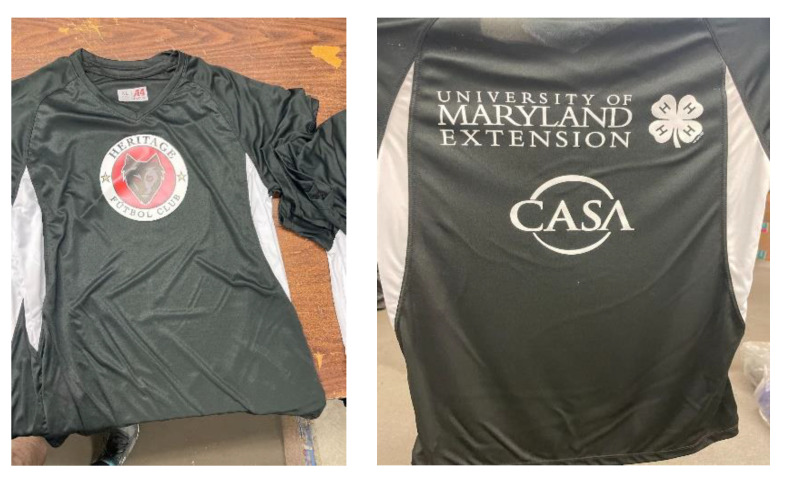
Custom-designed soccer jersey (front and back).

**Figure 4 ijerph-20-00207-f004:**
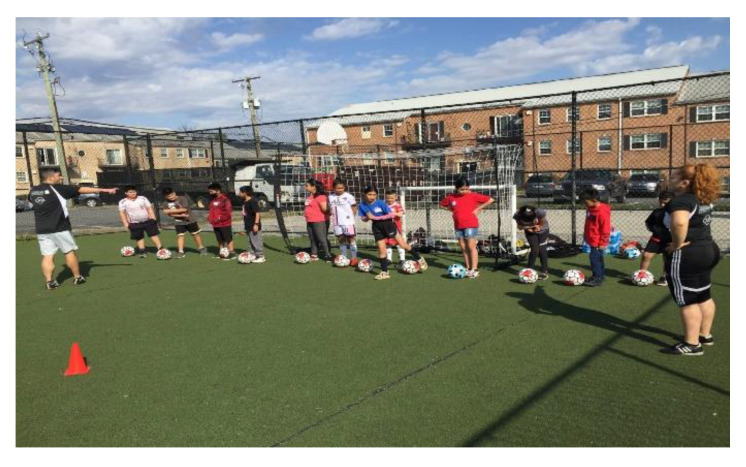
Youth soccer session.

**Figure 5 ijerph-20-00207-f005:**
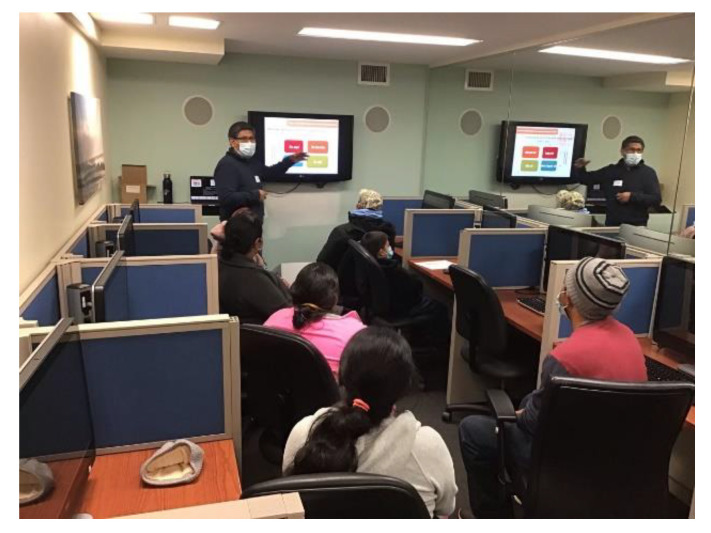
Parents session.

**Table 1 ijerph-20-00207-t001:** A list of the topics by week in the parent sessions.

Session	Topic
1	Parenting practices and healthy habits
2	Multiple cultures, living an active lifestyle
3	Adolescent development and healthy foods
4	Communication and limiting screen time
5	Rules, expectations, and healthy beverages
6	Managing conflicts and healthy snacks
7	Monitoring/supervision and fast food
8	Connecting with your child and family meals

## Data Availability

Data might be available upon reasonable request from the corresponding author.

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
