# Peer review of "The Development and Pilot Evaluation of a Family-Based Education to Strengthen Latinx Adolescent Mental Health in the United States: The Familias Activas Experience"

_ijerph, 2022, doi:10.3390/ijerph20010207_

Round 1
Reviewer 1 Report
Thank you for the opportunity to review the manuscript "The development and pilot evaluation of a family-based education 2 to strengthen Latinx adolescent mental health in the United States: 3 The Familias Activas experience." In this manuscript, the authors describe the development of a strength-based culturally safe intervention aimed at a Latinx community in X and present some findings from a pilot evaluation. I want to congratulate the authors on the project. It is a great initiative, and I wish you further success in later iterations of the project.
Please, find below some suggestions for the manuscript:
Introduction
- The intervention aims to strengthen mental health and lower anxiety amongst Latinx youth. However, the introduction only has one line (Lines 44-45) about poorer mental health outcomes in this population. I suggest adding more information about this issue in the introduction (perhaps even replacing the line about higher rates of unintended pregnancy and STDs with this further information?). This suggestion is further supported by Line 90, which claims that the introduction detailed the health challenges faced by this population.
- The last paragraph of the introduction mentions things that had not yet been presented to the reader (e.g., "the major goals of the project," "PYD programs"). I suggest explaining that these things will be discussed later (like a description of the following sections). Another possibility, not mutually exclusive, is to focus this section on showing the importance of addressing mental health in this population and finishing with a line about the overall goal of this manuscript.
Positive Youth Development
- Line 65 talks about focusing on youth's potential rather than their incapacities. I suggest rewording this phrase to supposed incapacities (as worded by Damon)
- Line 91 explains that, with effective sports-based programs, Latinx youth can succeed. What do the authors mean by succeeding?
- Would the authors consider it a good idea to move the lines starting at Line 92 ("Moreover, …") and finishing at Line 96 ("… programs.”) to the introduction?
- Line 99 explains that "cultural needs" are related to offering services in one's preferred language and targeting one's specific health needs. While it is important for services to address an individual's specific health needs, I would encourage the authors to add a bit more about cultural needs beyond language (even if it is just one more phrase between commas). The authors do this in the following lines, but if someone were to make a direct quote of this sentence, it might seem like using someone's preferred language is enough to address people's cultural needs.
Study objective
- In Line 120, the authors explain that they were engaging, especially fathers. Was there a particular reason to do this?
- It looked like part of the first short-term goals were to engage with families and provide family life education. Was this not the case?
- Line 131. Is the line "sugar-sweetened beverages" missing something like "reducing"? (keep in mind that I speak English as a second language)
- In Line 134, the authors explain that the family-skills component is part of "Padres Preparados." Was the sports-based component the newest component of a more extensive program (perhaps first intended to do this at a second stage?)? Was this why the first short-term goals were focused on youth and not parents? Could the authors elaborate on the relationship between these programs?
Materials and Methods
- The authors describe three teams: the collaborative team, the program delivery team, and the core research team. Perhaps introducing this team more explicitly would be helpful for readers. Also, did all teams meet weekly?
Curriculum Development
- In Line 196, the authors describe their website and mobile application. Could you please provide the reader with further information about how these were developed or the timing in which they were developed (e.g., in parallel, previously, as part of Padres Preparados)? It might not be necessary to add a lot of information. Still, I am thinking about people who might want to replicate the project (they might find this additional helpful information).
Structure of Youth and Parent Sessions
- Youth sessions lasted one hour, but parents' sessions lasted four hours. Could you please explain to the readers if these sessions were run in parallel? What were the young people doing for the remaining three hours if they were run in parallel? (e.g., they went home if there was someone that could look after them, and they were supervised during these three hours)
- Line 263, this sentence might also be better placed in the introduction.
Measures
- I couldn't find a reference to the "kid screen for parents" or the questions in the reference provided (reference number 10).
Results
- The authors mention items like LS-Y12, LS-Y15, KS-P3, and KS-P4. It might be helpful to describe these items to the reader further.
- Would the authors explain to the reader what 4-H is?
- I couldn't find a reference to the measures Kid screen for youth & Resilience in the results. Could the authors explain if participants did not complete these scales?
Discussion
- Line 406 mentions systematic reviews, but a reference to these systematic reviews is missing.
- The sentence starting at Line 409 mentions that "sports are linked to psychosocial" without clarifying psychosocial what exactly.
- In Line 413, the authors explain that the program evaluation had overall positive scores. Would it be more accurate to say that there were overall positive scores for the program's quality?
Conclusion
The authors explain that the feedback will influence the ongoing development of the program. However, this is not discussed in the discussion. Are the authors referring to the scales used? Are there any others changes that may be implemented based on feedback? This information may be particularly interesting for people considering replicating the project.
Author Response
Thanks for all your suggestions. Please find attached the responses to your suggestions in red in the document.

Reviewer 2 Report
Thank you for the opportunity to review this manuscript. Low-income populations usually suffer problems to reach the same opportunities than middle and high-income populations. Furthermore, it is well-known that this type of population often suffer mental health problems from early ages. For these reasons, it has been a pleasure to review this paper. However, some issues should be addressed to ensure the quality of the article. Below, authors can find some suggestions:
Abstract: The abstract does not show the main results. many times, authors write “the characteristics are explained throughout the manuscript”. However, when readers (like me) read the abstract, we hope to find the most important ideas of the paper. Please, modify the abstract to get the abstract’s aim.
- Line 21: “In this paper”. Review, please
Introduction:
- Line 44: “STDs” Explain before, please
- What is the social situation of Latinx population? Which are the differences among other ethnics? Is it the same that vulnerable population? I suggest describing better the Latinx’s characteristics.
- I suggest explaining better the concept of PYD
- Lines 76 to 87: Several PYDs programmes are explained throughout these lines, however the target population of these study is not Latinx people. I would be better to focus on PYDs programmes conducted among Latinx population.
- The last paragraphs are aimed at describing Latinx’s characteristics. Please, change the order of the introduction.
Materials and methods: The information of the sections is mixed. I suggest re-organizing it.
- Line 174: Describe better the characteristics of the families (e.g., ages, gender, number of children, incomes, ethnicity, etc.)
- Program’s design: Authors commented before the design of the programme was based on CbPAR (participatory action research). In this section, it is usually to describe the design and justify it. However, the information of this points seems confuse
- Suddenly, authors say they used a theory of change. If it is true, some information about this theory should be showed over the introduction.
- What happen if families do not have access to the internet due to their low-incomes?
- I believe that sessions should be based on effective strategies which have been proved before. It is needed more information about: what type of strategies have been development, why authors have decided used them, why authors have decided to implement soccer sessions instead of other sports, who decide how many sessions are developed over week, why father instead of mothers.
I think the program address too much information and superficial information. It is needed to dig deeper to understand the program.
Recently, in this journal has been published a paper which describes in detail a intervention programme (https://www.mdpi.com/1660-4601/19/11/6830) .
- Line 315: Remove reliability or complete, please
- Measures: Maybe it is better to create a table. It is weird to see the information in that format
Author Response
Thanks for all your suggestions. Please find attached the responses to your suggestions in red.

Round 2
Reviewer 2 Report
The quality of the manuscript continues poor
Author Response
Arlington, VA, USA
December 11, 2022
International Journal of Environmental Research and Public Health
Special issue: Issue "Children’s Mental Health, Parenting, Family and Groups’ Resilience in Crisis"
Dear Editors and Reviewers:
We would like to submit the updated version of our manuscript “The development and pilot evaluation of a family-based education to strengthen Latinx adolescent mental health: The Familias Activas experience” to the International Journal of Environmental Research and Public Health, special edition issue: "Children’s Mental Health, Parenting, Family and Groups’ Resilience in Crisis"
Considering the second round of reviews, we have made changes on the manuscript to make it stronger. The changes are the following:
- In the section 1.1 Positive Youth Development, we deleted the reference [16] of Sanders and colleagues so this section would be more focus on PYD programs in the United States.
- In the study objective we have been clearer providing explanation for the three components of the program 1) family training 2) sports 3) mental health.
- We have also improved the wording of the section Study Design and Conceptualization making clearer how these three components took place during the pilot. For example, we further explained that the training portion of the program comes from a previous successful educational program called Padres Preparados, Jovenes Informados.
- We moved point 2.3 Curriculum development to the Study objective section since we think this section is better served in this part. Having the graph of the pathway in the Study objective section will help our readers to better grasp the notion of the program since the reader can visually see the three main components of our project and what is the ultimate outcome, we want to achieve which is improve youth mental health.
- We have updated the Reference page and made sure they are relevant to our study.
- Considering the recommendation of having our Measures as tables. We reviewed other articles related to public health or health communication and found ‘Measures’ are usually included in the texts and not in tables. Here are some examples of journal articles. These are not open access articles, so we can submit one sample through the Journal portal.
Examples:
https://www.jstor.org/stable/48509721
https://www.tandfonline.com/doi/abs/10.1080/10410230902805932
https://pubmed.ncbi.nlm.nih.gov/26861963/
Thanks for your consideration,
The authors
